# Turn That Frown Upside Down: FaceID Customization via Cross-Training Data

## Abstract

Existing face identity (FaceID) customization methods perform well but are limited to generating identical faces as the input, while in real-world applications, users often desire images of the same person but with variations, such as different expressions (e.g., smiling, angry) or angles (e.g., side profile). This limitation arises from the lack of datasets with controlled input-output facial variations, restricting models' ability to learn effective modifications.

To address this issue, we propose CrossFaceID, the first large-scale, high-quality, and publicly available dataset specifically designed to improve the facial modification capabilities of FaceID customization models. Specifically, CrossFaceID consists of 40,000 text-image pairs from approximately 2,000 persons, with each person represented by around 20 images showcasing diverse facial attributes such as poses, expressions, angles, and adornments. During the training stage, a specific face of a person is used as input, and the FaceID customization model is forced to generate another image of the same person but with altered facial features. This allows the FaceID customization model to acquire the ability to personalize and modify known facial features during the inference stage. Experiments show that models fine-tuned on the CrossFaceID dataset retain its performance in preserving FaceID fidelity while significantly improving its face customization capabilities.

To facilitate further advancements in the FaceID customization field, our code, constructed datasets, and trained models are fully available to the public.

## 1 Introduction

Face identity (FaceID) customization is an important image generation task (Ramesh et al., 2022; Saharia et al., 2022; Rombach et al., 2022; Gal et al., 2022; Kumari et al., 2023; Ruiz et al., 2023; Xu et al., 2024), allowing users to achieve personalized facial customization using a pre-trained text-to-image diffusion model. Although existing methods demonstrate effectiveness, they exhibit a significant limitation: they can only generate images with the exactly same face as the input, while in real-world applications, users often desire images of the same individual but with variations, such as different facial expressions (e.g., smiling) or angles (e.g., side profile), shown in Figure 1.

This issue is primarily due to the lack of such a FaceID customization dataset where input and output faces exhibit controlled variations. In current datasets for training face customization models, the input and output faces in the dataset are often identical (Kim et al., 2022; Chen et al., 2023; Zhao et al., 2023; Peng et al., 2024). This setup restricts the model's ability to learn effective facial feature modifications during training, instead reinforcing its focus on maintaining face consistency. Consequently, the model struggles during inference when it is required to modify facial features while preserving FaceID consistency due to the lack of relevant training experience.

To address the above issue, in this paper, we propose CrossFaceID, the first large-scale, high-quality, and publicly available dataset specifically designed to improve the facial modification capabilities of FaceID customization models. Specifically, to obtain multiple public images of the same person, CrossFaceID gathered 40,000 images from approximately 2,000 celebrities, with each celebrity represented by around 20 images showcasing diverse facial attributes such as poses, expressions, angles, and adornments. To annotate these images, we use GPT-4 to generate detailed descriptions for these 40,000 images, particularly focusing on the facial features of the individuals, resulting in a set of 40,000 one-to-one text-image pairs.

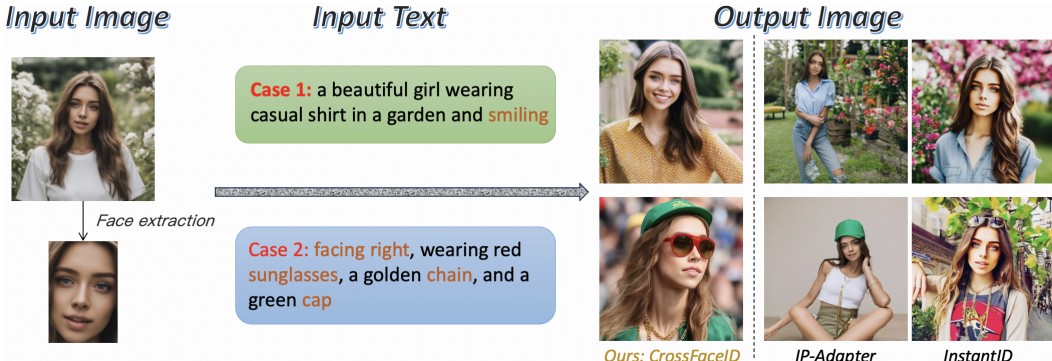

Figure 1: An example illustrating the limitations of existing FaceID customization models in generating images of the same individual with variations. Here, we use the same image of a girl as the input, paired with two different text prompts. In **Case 1**, the desired output is an image of the person with smiling, and in **Case 2**, the goal is an image of the person facing right and wearing sunglasses. The results clearly show that the two leading FaceID customization models, IP-Adapter (Ye et al., 2023) and InstantID (Wang et al., 2024), perform well in preserving the exact same face as the input image but fail to customize the input face as specified, such as adding a smile for Case 1 or generating a right-facing person with sunglasses for Case 2. In contrast, the model trained on our proposed CrossFaceID dataset effectively addresses these shortcomings, successfully generating a smiling face for Case 1 and a right-facing person wearing sunglasses for Case 2.

In this way, during the training stage, we start with a pre-trained FaceID model to ensure a solid foundation for preserving FaceID fidelity. Then, we employ a cross-training method, where a specific face of a person is used as input, and the pre-trained model is forced to generate another image of the same person but with altered facial features. This allows the pre-trained FaceID model to acquire the ability to personalize and modify known facial features during the fine-tuning process.

Trained on our CrossFaceID dataset, we achieve comparable performance in preserving FaceID fidelity when compared to the widely used FaceID frameworks, InstantID and IP-Adapter, while also significantly enhancing their ability to customize FaceID. In short, our contributions are:

1 We collect CrossFaceID, a high-quality, publicly available dataset designed to improve the facial modification capabilities of FaceID customization models.

2 The model fine-tuned on the CrossFaceID dataset retains its performance in preserving FaceID fidelity while significantly improving its face customization capabilities, showing the effectiveness of the dataset.

3 Our code, models, and datasets are fully available to the public, supporting further advancements in the FaceID customization field.

## 2 RELATED WORK

**Text-to-image Models.** Text-to-image refers to the process of generating images from textual descriptions using pre-trained image generation models (Ramesh et al., 2021; Ding et al., 2021; 2022; Ramesh et al., 2022; Rombach et al., 2022; Saharia et al., 2022; Huang et al., 2023). These models are trained to understand the relationship between textual input and visual content, enabling them to create images that match the given description. Thanks to the success of the transformer model (Vaswani, 2017), most early text-to-image approaches can be broken down into two stages: (1) using an image encoder, such as DARN (Gregor et al., 2014), PixelCNN (Van den Oord et al., 2016), PixelVAE (Gulrajani et al., 2016), or VQ-VAE (Van Den Oord et al., 2017), to convert an image into several tokens; and (2) training the model to predict these image tokens based on the provided text input within the transformer framework (Vaswani, 2017). Recently, diffusion models (Song et al., 2020a;b; Nichol et al., 2021; Dhariwal & Nichol, 2021; Ramesh et al., 2022; Saharia et al., 2022; Rombach et al., 2022; Balaji et al., 2022; Huang et al., 2023) have emerged as the new state-of-the-art approach for image generation, offering innovative solutions for the text-to-image task. In this

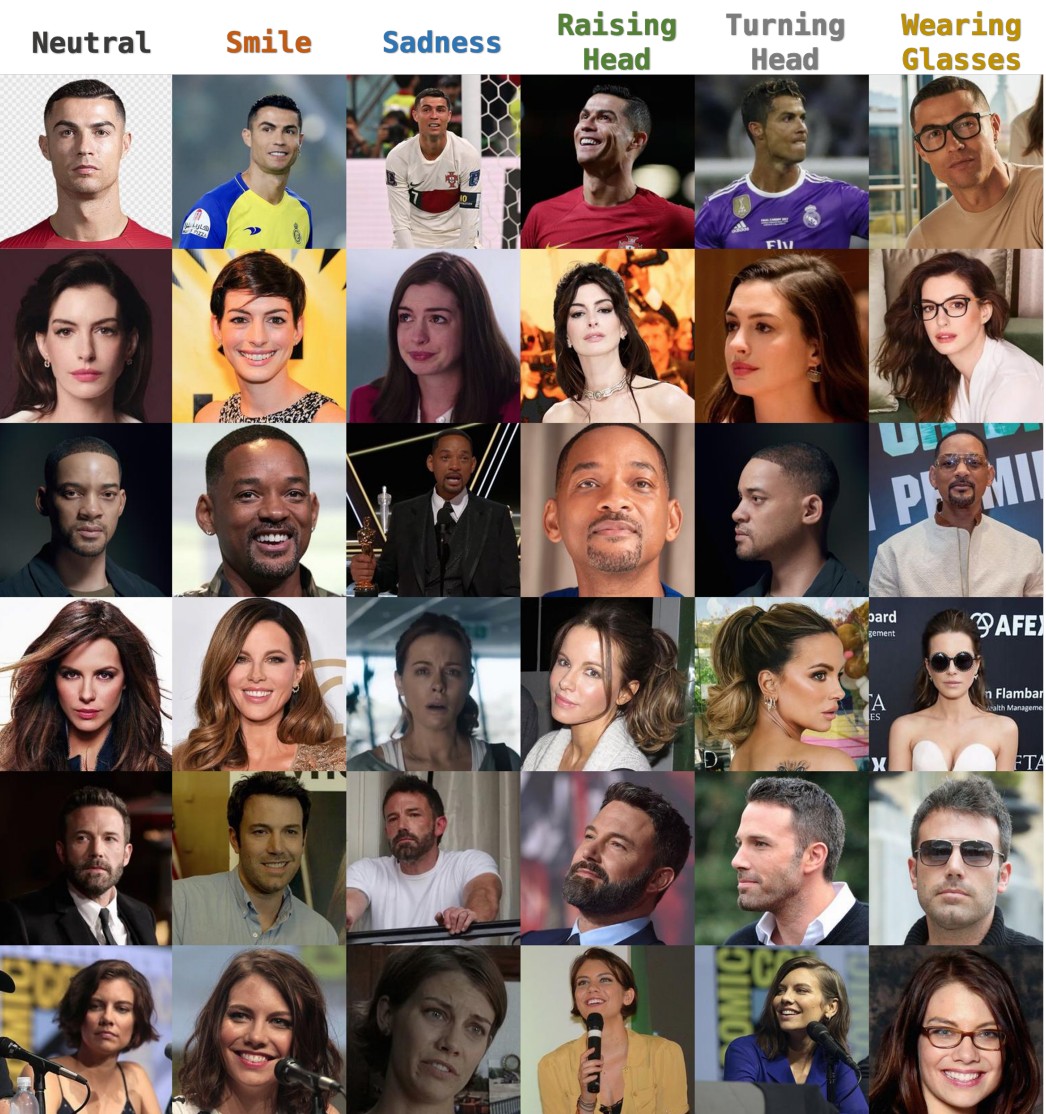

Figure 2: Images sampled from our constructed CrossFaceID dataset are presented. From top to bottom, the display includes six persons, each occupying one row with six images showing different expressions and angles: **(1) No Expression**: a straight-faced person with no expression, **(2) Smile**: a person with a smiling face, **(3) Sad**: a person with a sad expression, **(4) Rise**: a person with his or her face rising up, **(5) Side**: a person with his or her face turned to one side, and **(6) Wearing Glasses**: a person wearing glasses.

approach, the text prompt is first encoded into embeddings using a pre-trained language model such as T5 (Raffel et al., 2020) or CLIP (Radford et al., 2021), and then these encoded embeddings are used to guide the diffusion process, resulting in the generation of high-quality images. For example, GLIDE (Nichol et al., 2021) employs a cascaded diffusion architecture with CLIP as the text encoder to condition on natural language descriptions, facilitating both image generation and editing. Imagen (Saharia et al., 2022) adopts T5, a generic large language model pre-trained on text-only corpora, as the text encoder of diffusion models, to further enhance the text understanding.

**ID Customization.** FaceID customization for text-to-image models refers to the process of personalizing the text-to-image generation model by tailoring it to better recognize, and generate facial features and attributes specific to individual users (Valevski et al., 2023; Ye et al., 2023; Yuan et al.,

2023; Chen et al., 2024; Wang et al., 2024; Xiao et al., 2024; Li et al., 2024; Peng et al., 2024). Most of these works are optimization-free methods, which directly encode FaceID information into the generation process. For instance, Face0 (Valevski et al., 2023) substitutes the last three text tokens with the projected face embedding in the CLIP space, using the resulting combined embedding to guide the diffusion process. In a similar vein, PhotoMaker (Li et al., 2024) adopts a related approach but enhances its ability to extract FaceID embeddings by fine-tuning specific Transformer (Vaswani, 2017) layers in the image encoder and merging the class and image embeddings. Additionally, IP-Adapter (Ye et al., 2023) and InstantID (Wang et al., 2024) leverage FaceID embeddings from a face recognition model rather than CLIP image embeddings, ensuring consistent ID representation. However, these methods primarily concentrate on improving FaceID fidelity while overlooking customization. As illustrated in Figure 1, due to the complex architecture required to preserve FaceID, it is challenging for them to customize the generated face simply by modifying the input prompt. For example, generating an image of a person smiling when the input face does not show a smile becomes difficult. In this paper, we address this issue by altering only the composition of the training data, without changing the model structure. Extensive experiments demonstrate that our approach can effectively modify or preserve the input FaceID based on the input text prompt.

## 3 DATASET CONSTRUCTION: CROSSFACEID

In this section, we provide details on the dataset construction of CrossFaceID, which aims to address the issue in existing FaceID customization datasets, where input and output faces do not exhibit controlled variations. The construction process starts with images of the same individual. Next, we gather multiple images that showcase the person under varying facial attributes, such as different expressions, angles, or adornments. Using GPT-4o (Hurst et al., 2024), we then generate detailed textual descriptions for each image, particularly focusing on the facial features of the individual, resulting in a dataset of one-to-one text-image pairs.

### 3.1 IMAGE COLLECTION

To gather multiple public images of the same individual, we focus on celebrities, as they have numerous publicly available images on the Internet showcasing variations in facial features. As a result, we crawled approximately 60K images from 1626 celebritys.

### 3.2 IMAGE FILTERING

The collected images from the Internet may include noise, such as blurred faces or low resolution, which could significantly impact the quality of the subsequent FaceID customization models. To address this issue, we manually established rules to carefully select high-quality images suitable for FaceID customization training:

**(1) The image must include faces, with a maximum of three faces allowed.** This condition is based on two key reasons: (1) when an image contains multiple faces, the model may have difficulty identifying which face to focus on, potentially mixing up facial features between individuals. (2) During the inference, the trained FaceID customization model is typically designed to work with a single primary face. Limiting the number of faces ensures consistency with the inference process.

**(2) The image resolution must be at least 512x512 or higher.** High-resolution images generally contain finer facial details, such as subtle expressions, skin texture, and small features like wrinkles or dimples. As a result, they offer richer visual information for the FaceID customization model to analyze, leading to improved feature extraction and more effective learning.

**(3) The face should be at least 4% of the image.** This is because larger facial regions provide more pixels dedicated to facial features, enabling the FaceID customization model to better capture details such as expressions, and facial textures, which are critical for FaceID customization. As for the number "4%", it was determined through iterative refinements and validated via human and model evaluations.

The detailed statistics for the remaining dataset are shown in Table 4. After cleaning, we ensure that the collected images include faces that are clear and of adequate size. Examples of cleaned images are shown in Figure 2.

### 3.3 Image Annotations with GPT-4o

Since the crawled images lack captions, which are essential for training FaceID customization models, we leverage GPT-4o to annotate these images. This annotation process generate detailed textual descriptions with a specific emphasis on the individual's facial features.

Given a person image, we prompt GPT-4o to generate detailed textual descriptions. The designed prompt can be decomposed into the following three aspects:

(1) Task overview,

"*Your goal is to using less than 4 fluent sentences to generate short, descriptive captions for this image.*"

which emphasizes that the task is an annotation task, and the generated captions should be concise and not overly lengthy to avoid negatively impacting the latter training process.

(2) Instructions on face description,

"*Note that your description should be as detailed as possible to describe the features of the face in the image.*"

which instructs GPT-4o to focus on providing detailed descriptions of the facial features in the image within the constraints of the output length. It should avoid elaborating extensively on non-essential background information, such as character actions, attire, or surroundings, as these are not central to our following FaceID customization task.

(3) Precautions,

"*Please note that your answer should not include any tabulation format.*"

which prevents GPT-4 from generating structured descriptions. This is due to our observations in experiments where GPT-4 tended to create structured outputs. The structured output provides a quick overview of the image's content, but it is not ideal for training FaceID customization models. During inference, users are more likely to provide free-form descriptions rather than structured text. Training with structured descriptions could make it difficult for the model to effectively interpret unstructured, natural language inputs.

Then, we use the above designed prompt to interact with GPT-4o for annotating the crawled images. The parameters for this process are configured as follows:

*model = gpt-4o, temperature = 0, top_p = 1, presence_penalty = 0, max_token = 4096*

To evaluate the relevance between the collected images and their annotated texts, we use the image-text matching (ITM) score, following the approach in Dai et al. (2024), to quantify alignment across the two modalities. Since no existing dataset directly corresponds to ours, we compare it with several related text–image datasets created for face detection or text-to-image generation. As shown in Table 3, the annotations generated by GPT-4 demonstrate strong alignment with the original images. In addition, Figure 5 illustrates the distribution of various facial attributes (e.g., smiling, neutral, wearing glasses) within our CrossFaceID dataset. In the next section, we describe how these data are used to fine-tune FaceID customization models and to run inference with the trained model.

## 4 CrossFaceID Based FaceID Customization

In this section, we delve into the training and inference processes utilizing our constructed Cross-FaceID dataset.

### 4.1 Training

During the training phase, we follow the training structure of previous FaceID customization strategies (Ye et al., 2023; Wang et al., 2024), while modifying the arrangement of the input and output.

Formally, supposed that the collected dataset includes $N$ persons, with each person represented by $n$ triples $(y_{image}, y_{text}, y_{face})$ triples, where $y_{image}$ denotes an image of the person, $y_{text}$ represents the corresponding text description, and $y_{face}$ refers to the extracted face from $y_{image}$. Firstly, we

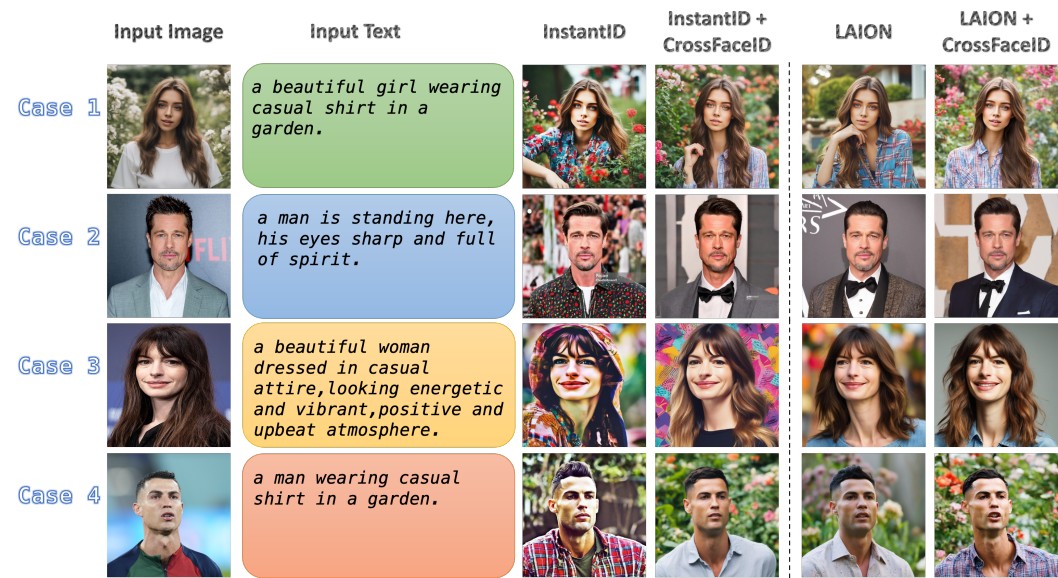

Figure 3: The results demonstrate the performance of FaceID customization models in maintaining FaceID fidelity. These results indicate that (1) for both the official InstantID model and the LAION-trained model, the ability to maintain FaceID fidelity remains consistent before and after fine-tuning on our CrossFaceID dataset, and (2) the model trained on our curated LAION dataset achieves comparable performance to the official InstantID model in preserving FaceID fidelity.

use one image of a person, $y^i_{image}, 0 \leq i \leq n$, as the input image $x_0$, and its corresponding text description $y^i_{text}$ as the text condition $C_{text}$. However, for the face condition $C_{id}$, we do not use its corresponding face $y^i_{face}$. Instead, we select a different random face of the same person, $y^j_{face}$, where $j \neq i$. Then, the subsequent training step follows the same procedure as standard FaceID customization models. Noise $\epsilon$ is sampled and added to the input image $x_0$ according to a predefined noise schedule (e.g., Gaussian noise), resulting in a noisy sample $x_t$ at timestep $t$. The diffusion model $\epsilon_\theta$ is then trained to predict the normally distributed noise $\epsilon$ using the current noisy image $x_t$, the timestep $t$, the text condition $C_{text}$, and the face condition $C_{id}$. For optimization process:

$$\mathcal{L}(\theta) = \mathbb{E}_{x_0, C_{text}, C_{id}, t, \epsilon \sim \mathcal{N}(0, \mathbf{I})} \| \epsilon - \epsilon_\theta(x_t, C_{text}, C_{id}, t) \|^2 \quad (1)$$

where $t \in [0, T]$ is the sampled diffusion step.

For settings, we follow the approach outlined in Wang et al. (2024) focusing on single-person images and utilizing a pre-trained face model, Antelopev2[1], to detect and extract face ID embeddings from human images. During training, only the parameters of the Image Adapter and IdentityNet are updated, while the pre-trained text-to-image model remains frozen. Our experiments are conducted using the SDXL-1.0 model on 16 NVIDIA H800 GPUs (80GB) with a batch size of 2 per GPU.

## 4.2 INFERENCE

The inference process follows the same approach as the diffusion models. It begins with a sample of Gaussian noise, represented by $x_T$, where $T$ is a predefined number of timesteps. This initial state, composed entirely of unstructured noise, serves as the starting point, representing a meaningless input image. At each timestep $t$, the model takes the noisy image $x_t$ as the input and utilizes the text prompt condition $C_{text}$ and the input face condition $C_{id}$ to predict the clean image or the noise that should be removed, progressively refining the image towards the final clean output $x_0$. The predicted noise $\epsilon_\theta$ is then used to update the noisy image, denoising step by step:

$$x_{t-1} = \alpha_t x_t - \sigma_t \epsilon_\theta(x_t, C_{text}, C_{id}, t) \quad (2)$$

---

[1] https://github.com/deepinsight/insightface

where $\alpha_t$ and $\sigma_t$ are two coefficients controlling the denoising process. Over several timesteps $T$, the noise is gradually removed, ultimately producing a customized, clean image.

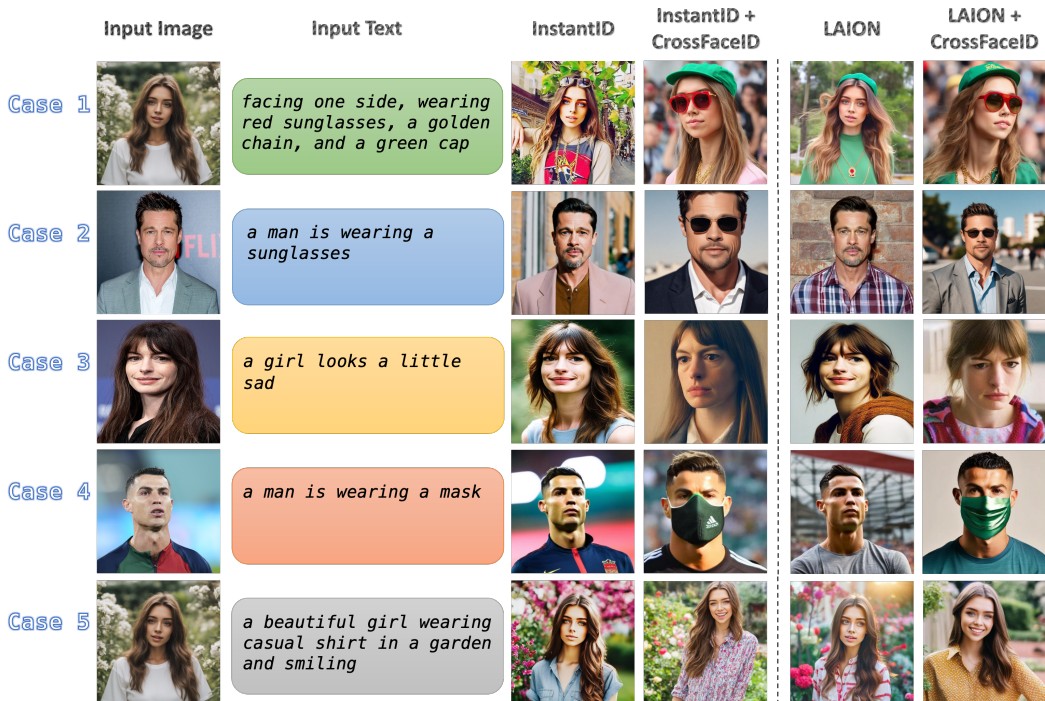

Figure 4: The results of the performance for FaceID customization models in customizing or editing FaceID. From these results, we can clearly observe an improvement in the models' ability to customize FaceID after being fine-tuned on our constructed CrossFaceID dataset.

## 5 EXPERIMENTS

To assess the effectiveness of CrossFaceID, we train the state-of-the-art FaceID customization model, InstantID (Wang et al., 2024), as well as its original version, IP-Adapter (Ye et al., 2023), using our constructed CrossFaceID dataset. We then perform comparative experiments to evaluate both FaceID fidelity and FaceID customization capabilities.

### 5.1 MAIN RESULTS

In this section, we conduct experiments to separately evaluate the FaceID fidelity and FaceID customization capabilities of models trained on our CrossFaceID dataset. For clarity, we refer to the official InstantID model as "InstantID," and the model further fine-tuned on our CrossFaceID dataset as "InstantID + CrossFaceID." Similarly, the InstantID model pre-trained on our curated LAION dataset is referred to as "LAION," while the model further trained on our CrossFaceID dataset is called "LAION + CrossFaceID." Below are results in FaceID fidelity and FaceID customization.

#### 5.1.1 FACEID FIDELITY

Figure 3 illustrates the performance of FaceID customization models in maintaining FaceID fidelity. From these results, we can conclude that (1) for both the official InstantID model and the LAION-trained model, the ability to maintain FaceID fidelity remains consistent before and after fine-tuning on our CrossFaceID dataset. Such as the case 1, all models, including the two fine-tuned on the CrossFaceID dataset generate the exact same girl as the input image and wearing shirt in a garden as the input text "a beautiful girl wearing casual shirt in a garden". This demonstrates that our constructed CrossFaceID dataset does not compromise the FaceID fidelity performance of these

FaceID customization models. (2) The model trained on our curated LAION dataset demonstrates performance comparable to the official InstantID model in maintaining FaceID fidelity. For instance, in case 2, both the official InstantID model and the LAION-trained model successfully generate the desired images based on the input. This ensures the fairness of our experiments when further fine-tuning CrossFaceID on models with comparable baseline performance.

### 5.1.2 FACEID CUSTOMIZATION

Figure 4 demonstrates the performance for FaceID models in customizing or editing FaceID. From these results, we can clearly observe an improvement in the models' ability to customize FaceID after being fine-tuned on our CrossFaceID dataset. For example, in case 3, although the input image features a smiling girl, both CrossFaceID models, "InstantID + CrossFaceID" and "LAION + CrossFaceID," successfully generate images of the girl without a smile, appearing slightly sad as specified by the input text, "a girl looks a little sad." Moreover, the two CrossFaceID models effectively customize the input person by generating a man wearing sunglasses for case 2 ("a man is wearing sunglasses") and a man wearing a mask for case 4 ("a man is wearing a mask").

| Model | CrossFaceID-test | | | Unsplash-50 | | |
|---|---|---|---|---|---|---|
| | Face Sim↑ | CLIP-T↑ | CLIP-I↑ | Face Sim↑ | CLIP-T↑ | CLIP-I↑ |
| IP-Adapter (Ye et al., 2023) | 0.29 | 0.19 | 0.62 | 0.57 | 0.24 | 0.61 |
| InstantID (Wang et al., 2024) | 0.31 | 0.24 | 0.69 | 0.61 | 0.27 | 0.68 |
| LAION | 0.31 | 0.25 | 0.71 | 0.62 | 0.28 | 0.70 |
| *Ours (Fine-tuned on CrossFaceID)* | | | | | | |
| IP-Adapter + CrossFaceID *w/o* cross-training | 0.30 (+0.01) | 0.19 (+0.00) | 0.63 (+0.01) | 0.57 (+0.00) | 0.24 (+0.00) | 0.63 (+0.02) |
| IP-Adapter + CrossFaceID | 0.30 (+0.01) | 0.25 (+0.04) | 0.67 (+0.05) | 0.59 (+0.02) | 0.27 (+0.03) | 0.67 (+0.06) |
| InstantID + CrossFaceID *w/o* cross-training | 0.31 (+0.00) | 0.24 (+0.00) | 0.71 (+0.02) | 0.61 (+0.00) | 0.29 (+0.02) | 0.69 (+0.01) |
| InstantID + CrossFaceID | 0.31 (+0.00) | 0.30 (+0.06) | 0.76 (+0.07) | 0.62 (+0.01) | 0.32 (+0.05) | 0.74 (+0.06) |
| LAION + CrossFaceID *w/o* cross-training | 0.32 (+0.01) | 0.25 (+0.00) | 0.74 (+0.03) | 0.62 (+0.00) | 0.29 (+0.01) | 0.72 (+0.02) |
| LAION + CrossFaceID | **0.33 (+0.02)** | **0.31 (+0.06)** | **0.79 (+0.08)** | **0.63 (+0.01)** | **0.34 (+0.06)** | **0.75 (+0.05)** |

Table 1: Quantitative results of different FaceID models on CrossFaceID-test and Unsplash-50, and we highlight the highest score in bold.

| Model | Cusomization | Fidelity | Quality |
|---|---|---|---|
| | *Avg* | *Avg* | *Avg* |
| IP-Adapter (Ye et al., 2023) | 1.2 | 1.7 | 3.01 |
| InstantID (Wang et al., 2024) | 1.64 | 4.17 | 4.36 |
| LAION | 1.65 | 4.2 | 4.38 |
| *Ours (Fine-tuned on CrossFaceID)* | | | |
| IP-Adapter + CrossFaceID *w/o* cross-training | 1.22 | 1.68 | 3.00 |
| IP-Adapter + CrossFaceID | 2.95 | 1.62 | 2.98 |
| InstantID + CrossFaceID *w/o* cross-training | 1.76 | 4.18 | 4.38 |
| InstantID + CrossFaceID | 4.02 | 4.17 | 4.40 |
| LAION + CrossFaceID *w/o* cross-training | 1.81 | 4.2 | 4.42 |
| LAION + CrossFaceID | **4.21** | **4.21** | **4.43** |

Table 2: Human evaluations of different FaceID customization models based on three criteria: (1) Customization, (2) Fidelity and (3) Quality, and we highlight the highest score in bold.

### 5.2 QUANTITATIVE RESULTS

To more effectively evaluate the effectiveness of our CrossFaceID dataset, we conduct quantitative experiments on two test sets: CrossFaceID-test and Unsplash-50 (Gal et al., 2024). Due to the lack of test sets for evaluating the abilities of models in customizing FaceID, we collected CrossFaceID-test. CrossFaceID-test consists of 200 text-image pairs sourced from the Internet. For each image, we include a version of the same person with a different facial expression or angle, allowing us to assess the models' performance in generating reference images that align with the input text and given face. For Unsplash-50, it includes 50 text-image pairs, which can be utilized to quantify the models' performance in maintaining FaceID fidelity.

For evaluation metrics, we use : (1) Face Sim, which calculates the FaceID cosine similarity between the input face and the face extracted from the generated image, providing a direct estimate of the difference between the generated and input faces; (2) CLIP-T (Radford et al., 2021), which evaluates the model's ability to follow prompts; and (3) CLIP-I (Radford et al., 2021), which measures the CLIP image similarity between the original image and the image after FaceID insertion.

Results are shown in Table 1. From these results, we can observe that our CrossFaceID dataset leads to consistent improvements across all metrics. For example, on the CrossFaceID-test set, the CLIP-I rises from 0.71 (LAION) to 0.79 (LAION + CrossFaceID), while on the Unsplash-50 dataset, the CLIP-T increases from 0.28 to 0.34. Furthermore, the results highlight the importance of cross-training: without it, the trained models lose the ability to edit or customize the input face. Specifically, for the "LAION + CrossFaceID" model, the CLIP-I drops from 0.79 to 0.74 on CrossFaceID-test, and the CLIP-T decreases from 0.34 to 0.29 on Unsplash-50 when cross-training is removed. These findings confirm the effectiveness of our CrossFaceID dataset in both customizing faces according to user descriptions and preserving facial identity in the generated images.

### 5.3 HUMAN EVALUATIONS

While automated evaluations, as conducted above, effectively measure objective aspects like FaceID fidelity and prompt adherence, they fall short in assessing subjective qualities, such as whether the customized face accurately represents the requested attributes (e.g., expressions or angles) while maintaining resemblance to the input person. To address this limitation, we incorporate human evaluations into our experiments. In this way, we collected 200 celebrity faces from the Internet and manually designed prompts to force the evaluated FaceID models to generate images showing different expressions and angles (e.g., smile, sadness, turning head and wearing attire). The generated images are then evaluated by 10 human participants, who score them based on three criteria: (1) **Customization:** whether the generated image accurately follows the input prompt to customize the given face, (2) **Fidelity:** whether the generated image retains the identity of the input face, and (3) **Quality:** the overall quality of the generated image. The scores range from 0 to 5, with 0 indicating the poorest quality and 5 representing the highest quality.

The results are shown in Table 2. From these results, we can observe that: (1) In terms of customization, previous FaceID customization models demonstrate very limited customization capabilities, with average scores of only 1.2 for the IP-Adapter model and 1.64 for the InstantID model. However, after fine-tuning on our CrossFaceID dataset, their customization abilities improve significantly, such as an increase from 1.65 (LAION) to 4.21 (LAION + CrossFaceID). (2) Regarding fidelity, the ability to maintain FaceID fidelity remains stable before and after fine-tuning on our CrossFaceID dataset. For example, 4.17 (InstantID) vs. 4.17 (InstantID + CrossFaceID) and 4.2 (LAION) vs. 4.21 (LAION + CrossFaceID). (3) In terms of quality, the fine-tuning process on our CrossFaceID dataset does not degrade the quality of the generated images but instead slightly improves it, such as an increase from 4.38 (LAION) to 4.43 (LAION + CrossFaceID).

## 6 CONCLUSION

In this paper, we propose CrossFaceID, the first large-scale, high-quality, and publicly available dataset specifically designed to improve the facial modification capabilities of FaceID customization models. Specifically, CrossFaceID consists of 40,000 text-image pairs from approximately 2,000 persons, with each person represented by around 20 images showcasing diverse facial attributes such as poses, expressions, angles, and adornments. During the training stage, a specific face of a person is used as input, and the FaceID customization model is forced to generate another image of the same person but with altered facial features. This allows the FaceID customization model to acquire the ability to personalize and modify known facial features during the training process, thus improving its FaceID customization abilities during the later inference stage.

We perform comprehensive experiments to demonstrate the effectiveness of our CrossFaceID dataset, revealing that models fine-tuned on this dataset maintain their ability to preserve FaceID fidelity while significantly enhancing their face customization capabilities. Moreover, to support further progress in the FaceID customization domain, we have made our code, datasets, and models publicly available.

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

## A  APPENDIX

| Model | Min ITM↑ | Max ITM↑ | Mean ITM↑ |
|---|---|---|---|
| LAION-Face (Zheng et al., 2022) | 0 | 1 | 0.42 |
| Multi-Modal-CelebA-HQ (Xia et al., 2021) | 0 | 0.93 | 0.51 |
| FFHQ-Text (Zhou & Shimada, 2021) | 0 | 0.89 | 0.49 |
| FaceCaption-15M (Dai et al., 2024) | 0 | 1 | 0.66 |
| *Ours (CrossFaceID)* | | | |
| CrossFaceID | 0.08 | 1 | 0.74 |

Table 3: Image-text matching score.

| Statists | Number |
|---|---|
| Total Celebrities | 1626 |
| Total Images | 40596 |
| Average Images Per Celebrity | 24.9 |
| Medium Images Per Celebrity | 7 |
| Maximum Images Per Celebrity | 111 |
| Minimum Images Per Celebrity | 2 |

Table 4: The image statistics for the cleaned CrossFaceID dataset.

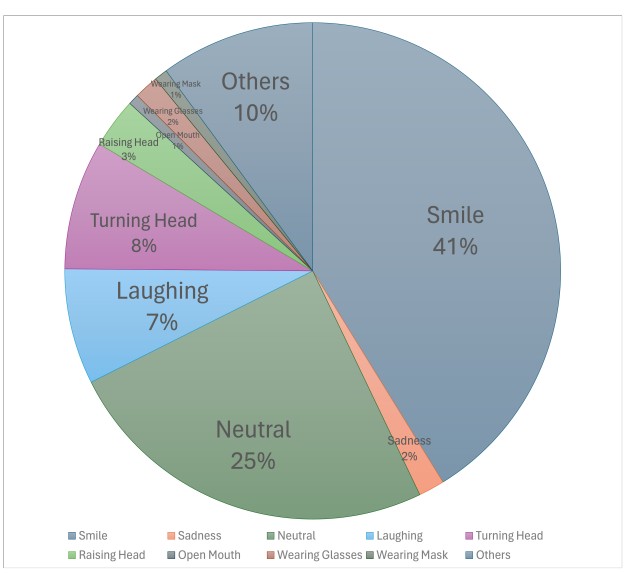

Figure 5: The distribution of various facial features (e.g., expressions and angles) within our Cross-FaceID dataset.

