# OpenReview forum: "Turn That Frown Upside Down: FaceID Customization via Cross-Training Data"
_ICLR.cc/2026/Conference — Submitted to ICLR 2026_

### Official Review · Reviewer_fcwF · 2025-10-29

**Soundness:** 2
**Presentation:** 1
**Contribution:** 2
**Rating:** 2
**Confidence:** 4

**Summary:**

This paper introduces CrossFaceID, a large-scale dataset containing 40,000 text–image pairs collected from approximately 2,000 individuals, with around 20 images per person. The dataset covers a wide range of facial attributes, including pose, expression, viewing angle, and accessories, and is designed to facilitate research on FaceID customization and facial editing.

However, the image authorization and copyright status are unclear, which raises serious concerns regarding data legality and ethical compliance. In addition, the annotations were not manually verified, suggesting potential issues with annotation accuracy and consistency.

**Strengths:**

1. The paper introduces a large-scale dataset (CrossFaceID) with 40,000 text–image pairs from 2,000 individuals, providing a valuable resource for training and evaluating FaceID customization models.

2. The dataset includes diverse facial variations (e.g., poses, expressions, angles, and accessories), which could help improve model generalization.

**Weaknesses:**

1. The images appear to lack explicit authorization, leading to major concerns about copyright and data usage rights. This issue may significantly limit the dataset’s public accessibility and research applicability.

2. The annotations were not manually verified, which raises doubts about the reliability and precision of the textual descriptions. Poor annotation quality could undermine the dataset’s effectiveness in training high-fidelity generative models.

3. The results shown in Figure 4 do not appear convincing. Some generated examples (e.g., Case 3 and Case 4) look unnatural and fail to demonstrate a clear advantage over existing methods.

**Questions:**

1. The images appear to lack explicit authorization, leading to major concerns about copyright and data usage rights. This issue may significantly limit the dataset’s public accessibility and research applicability.

2. The annotations were not manually verified, which raises doubts about the reliability and precision of the textual descriptions. Poor annotation quality could undermine the dataset’s effectiveness in training high-fidelity generative models.

3. The results shown in Figure 4 do not appear convincing. Some generated examples (e.g., Case 3 and Case 4) look unnatural and fail to demonstrate a clear advantage over existing methods.

**Details Of Ethics Concerns:**

This paper introduces a dataset containing 40,000 text–image pairs collected from approximately 2,000 persons.
The images appear to lack explicit authorization, leading to major concerns about copyright and data usage rights.

---

### Official Review · Reviewer_m2YN · 2025-10-30

**Soundness:** 3
**Presentation:** 3
**Contribution:** 2
**Rating:** 4
**Confidence:** 3

**Summary:**

This paper follows a data-centric approach and collects a dataset of 40,000 of text image pairs. They use the collected dataset to fine-tune existing models to generate diverse facial variations (expression and pose) of a given input identity.

**Strengths:**

- The collected dataset of image-text pairs for customization is valuable and can enable further research in this area.
- The qualitative results look realistic and show that they can indeed generate different expressions and head poses based on instructions
- The quantitative results of their fine-tuned models show consistent improvement over their base model counterparts and do not seem to degrade.

**Weaknesses:**

- There is limited technical novelty. Mostly just fine-tunes a model with a new dataset.
- The dataset only contains a very limited set of facial features.
- There's no stratified analysis, only aggregate results. Given the highly imbalanced facial features in the dataset, the customization performance results could be heavily dominated by just one feature that the model can do very well (e.g., smiling).
- Currently, there are no convincing experiments to validate the diversity. It would be nice to incorporate this, especially since fine-tuning can sometimes also have the unintended consequence of reduced diversity.
- InstantID also allows for novel view synthesis using a reference view. It would be nice to have a comparison with this and also show the diversity of head pose / angles that your model can generate as compared with InstantID.
- It would also be nice to see an experiment showing performance as a function of the number of fine tuning images.

**Questions:**

- How does the highly imbalanced distribution of facial features affect customization performance for each feature?
- How many examples of each feature do we need for fine tuning to capture a specific facial feature?
- Were the identities for fine-tuning different from the identities for testing? It was not explicitly clear in the paper.
- Also, are there visual examples showing performance for a bit more out-of-distribution faces, such as non-celebrity faces, other races, non front facing poses, face accessories and more extreme make ups? It would be good to see the model's robustness to these inputs.

**Details Of Ethics Concerns:**

They collect face images of approximately 2000 celebrities from the web.
Not sure if this necessitates an ethics review, but mentioning here just in case.

---

### Official Review · Reviewer_QtiN · 2025-11-02

**Soundness:** 2
**Presentation:** 3
**Contribution:** 2
**Rating:** 2
**Confidence:** 4

**Summary:**

This paper collects a dataset, called CrossFaceID, which is specifically designed to improve the facial modification capabilities of FaceID customization models. CrossFaceID consists of 40,000 text-image pairs of 2,000 persons. Based on the collected dataset, the authors design a cross-training strategy to fine-tune the FaceID customization model, allows it to acquire the ability to personalize and modify known facial features during the inference stage.

**Strengths:**

1. The paper is well structured and easy to follow. The construction process of the CrossFaceID dataset is introduced in detail.

2. Experimental results appear to demonstrate the effectiveness of the CrossFaceID dataset and the proposed cross-training strategy.

**Weaknesses:**

1. One of my major concerns lies in the main contribution of this paper. The authors claim that the CrossFaceID dataset is a large-scale and high-quality dataset, but it only contains 40,000 text-image pairs of 2,000 persons. Compared with existing text-to-image person generation datasets (e.g., CosmicMan), the data volume of CrossFaceID is relatively smaller. Moreover, the data collection process follows a standard practice, so I cannot get any novel insight from this dataset.

2. For the proposed cross-training strategy, the experiments in Figure 4 show that it can empower the model to generate the results that can more accurately align with the text prompt, such as “wearing a sunglass” in case 2 and “wearing a mask” in case 4. However, from the technical details of the cross-training strategy, I cannot find any specific mechanism to achieve this text-image alignment. I believe that it cannot be realized by just changing the ID prompt.

3. The motivation of this paper is relatively weak. The authors claim that their motivation is “Existing face identity (FaceID) customization methods perform well but are limited to generating identical faces as the input, while in real-world applications, users often desire images of the same person but with variations, such as different expressions (e.g., smiling, angry) or angles (e.g., side profile).” However, based on my knowledge, RealisID [1] is a scale-robust and fine-controllable ID customization method, which can generate face images with different facial expressions, head poses, face locations, face scales, and even multiple faces.

[1] Sun Z, Du F, Chen W, et al. RealisID: Scale-Robust and Fine-Controllable Identity Customization via Local and Global Complementation[C]//Proceedings of the AAAI Conference on Artificial Intelligence. 2025, 39(7): 7158-7166.

4. In the experiments, the authors pre-trained the competing methods by using “our curated LAION dataset”. How to construct this dataset? Is there any description and reference for this dataset?

**Questions:**

Please refer to Weaknesses.

**Details Of Ethics Concerns:**

The authors collect a dataset that contains 4,0000 face images from approximately 2,000 celebrities. Will it raise some concerns about human privacy?

---

### Meta-Review · Area_Chair_eJDt · 2026-01-06

**Summary:**

This paper received mixed and generally low scores in the initial review phase, including two scores of 2 and one score of 4. The main concerns raised by reviewers focused on limited technical novelty, weak motivation relative to existing FaceID customization methods, insufficient experimental analysis, unclear dataset construction details, and significant ethical and legal concerns related to data collection.

Crucially, **the authors did not submit a rebuttal**. As a result, none of the reviewers’ concerns were addressed or clarified during the rebuttal phase.

**Reviewer Concerns:**

No rebuttal was submitted by the authors. As a result, no reviewer concerns were addressed during the rebuttal phase. All concerns raised in the initial reviews remain outstanding.

**Reviewer Scores:**

| Reviewer | Initial Score | Estimated Final Score | Reason |
|--------|---------------|----------------------|--------|
| QtiN | 2 | 2 | Major novelty, motivation, and ethics concerns remain unaddressed |
| m2YN | 4 | 4 | Technical novelty and evaluation concerns not addressed |
| fcwF | 2 | 2 | Legal, ethical, and data quality issues unresolved |

---

### Decision · Program_Chairs · 2026-01-26

Reject